# OpenReview forum: "Composing Parameter-Efficient Modules with Arithmetic Operation"
_NeurIPS.cc/2023/Conference — NeurIPS 2023 poster_

### Official Review · Reviewer_hxSN · 2023-06-25

**Soundness:** 3 good
**Presentation:** 3 good
**Contribution:** 2 fair
**Rating:** 3
**Confidence:** 4

**Summary:**

This paper studies how to combine parameter-efficiently tuned models in the parameter space. The authors define two kinds of Arithmetic Operation for parameter ensemble: addition and negation. They evaluate their methods for distribution generalization, multi-tasking, detoxifying, and domain transfer. Extensive experiments are conducted to support the claims.

**Strengths:**

+ The paper is clear and easy to follow.

+ The paper studies an interesting topic, i.e., parameter ensemble for LLMs.

**Weaknesses:**

- The idea is not novel. Parameter ensemble (weight averaging) has been proposed for a while, and both addition and negation methods are included in previous works [1,2,3,4]. Especially, there is an important missing reference [1] that defines linear interpolation in the parameter space, which I believe is a more general method that encompasses both "addition" and "negation".

- Some explorations have already been conducted before, e.g., distribution generalization [1, 3], multi-tasking and detoxifying [1], and domain transfer [4]. I think the authors should explicitly discuss the difference and novel findings compared with existing literature.

- Although the authors study the effects of many aspects, the analysis is partial and doesn't reach saturation so hard to make strong deduction out of it. I think the authors should delve deeper into the superficial findings to better understand why parameter averaging brings these benefits.

Please correct me if I misunderstood anything.

[1] Rofin, Mark, Nikita Balagansky, and Daniil Gavrilov. "Linear Interpolation In Parameter Space is Good Enough for Fine-Tuned Language Models." arXiv preprint arXiv:2211.12092 (2022).

[2] Wortsman, Mitchell, et al. "Model soups: averaging weights of multiple fine-tuned models improves accuracy without increasing inference time." International Conference on Machine Learning. PMLR, 2022.

[3] Qin, Yujia, et al. "Exploring Mode Connectivity for Pre-trained Language Models." arXiv preprint arXiv:2210.14102 (2022).

[4] Ilharco, Gabriel, et al. "Editing Models with Task Arithmetic." arXiv preprint arXiv:2212.04089 (2022).

**Questions:**

NA

---

> ### Author Rebuttal · Authors · 2023-08-09
>
> We thank the reviewer for the time and comments. Nevertheless, we kindly disagree with the reviewer, especially on the “novelty” assessment. It seems that there might be some misconceptions regarding our work's motivations and its differentiation from prior studies. We aim to elucidate our motivations and underscore the novelty of our contributions from three perspectives:
> 1. *Motivation*: The primary drive behind this research is the composition of parameter-efficient modules as opposed to merely ensembling every parameter from fully fine-tuned models. We posit that the composition of PEMs stands apart as a research problem from the traditional full model ensembling, and we view PEM composition as a way towards modular deep learning, which is more flexible and presents a realm of intriguing future prospects, distinct from standard full model ensembling. Our work aims to establish a simple attempt towards the goal. Therefore, our paper makes unique contributions to the direction of PEM composition and modular deep learning, while most of the papers mentioned by the reviewer (most of them are already cited in the submission, and we will cite [4] for completeness) focus on ensembling full fine-tuned models.
>
> 2. *Method*: We are the *first* to design both addition and negation operators for PEM composition. While our addition operator takes the form of a simple average, our negation operators, especially the one for $\rm{(IA)^3}$, are not trivial to design and diverge from the one in [1,4] that cater to full parameter ensembling and just naively negates all the parameters in a linear interpolation scheme – our negation operators for both LoRA and $\rm{(IA)^3}$ (Eq 5 and Eq 6) do not trivially follow a linear interpolation formulation as in previous work. For example, in linear interpolation of [4], their formulation only covers negating ALL the parameters of the module and assigning a weight to it, and both Eq 5 and Eq 6 in our submission do not follow this formulation. In our theoretical explanation in the paper and empirical ablation results in the response to Q2 of Reviewer EMdb, we show that our negation operator design is necessary and naively negating the parameters with a linear interpolation formulation as in previous study is ineffective in PEM composition.
>
> 3. *Empirical Results*:
>     * The reviewer mentioned that our empirical explorations have been conducted before – we respectfully disagree. Few previous work [2,3] only conducted the simple parameter addition of PEM modules in limited settings, other papers mentioned by the reviewer such as [1] involve both addition and negation operators but is confined to full model ensembling – however, their conclusion and observations do not seamlessly transit to the PEM context, given that we have shown that their negation operation does not work effectively for composing PEMs. This solidifies our position as the *first* to introduce and scrutinize negation operators tailored for PEM composition and to assess the method in a range of scenarios.
>     * Particularly, we are the *first* to study PEM composition in instruction tuning based on modern LLMs such as LLaMA. This not only underscores our method's applicability but also its relevance to the challenges faced in today's LLM landscape.
>     * In previous work on ensembling parameters of full fine-tuned models, it is commonly believed that the parameters to be averaged need to stem from the same initialization (pretrained model checkpoint). Our paper explores the setting where this assumption does not hold in the LoRA experiment, thus we think that our empirical study on the initialization effect also provides new insights.
>
> [1] Ilharco, Gabriel, et al. "Editing models with task arithmetic." ICLR. 2022.
>
> [2] Qin, Yujia, et al. "Exploring Mode Connectivity for Pre-trained Language Models." EMNLP. 2022.
>
> [3] Ponti, Edoardo Maria, et al. "Combining Parameter-efficient Modules for Task-level Generalisation." EACL. 2023.
>
> [4] Rofin, Mark, Nikita Balagansky, and Daniil Gavrilov. "Linear interpolation in parameter space is good enough for fine-tuned language models." arXiv preprint arXiv (2022).

---

### Official Review · Reviewer_PsaT · 2023-07-02

**Soundness:** 3 good
**Presentation:** 3 good
**Contribution:** 3 good
**Rating:** 7
**Confidence:** 3

**Summary:**

The authors propose composing parameter-efficient modules (primarily LoRA and IA3 modules) directly in weight space, and show that simple linear combinations are able to achieve good performance in distributional generalization, multitasking, unlearning/de-toxifying, and domain transfer. The authors also show their approach works for both T5 and LLaMA models. Analysis of the approach suggests that their lora compositions are not as sensitive to different initializations as adapters.

**Strengths:**

Straight-forward method, and well-tested across a number of diverse settings (different modules, different tasks, different underlying models).
The paper is clear and well-structured, and the initialization experiments in the analysis section are interesting. I think the exploring merging of parameter-efficient modules is important and interesting work.


**Weaknesses:**

No glaring weaknesses. I think how lambda is chosen for tasks needs to be explained, unless I missed it - comparing Table 2 and Figure 2, it seems like lambda for Table 2 is > .5, so explaining how you chose the number is important. I see for the unlearning experiment you did tune lambda somewhat (Appendix A). If you have to tune lambda in all cases, this makes this approach more difficult to apply in practice due to the extra tuning required.

The domain transfer experiment in 4.5 is unclear to me - do you also train a module on the classification datasets to get theta^{amazon_cls/yelp_cls}? Looking at Figure 3, it seems lambda = 1.0 does best, which implies just using a trained classifier module directly with no composition, right?

Results are mostly over classification tasks - it would be interesting to see e.g. the multitasking-style experiments applied to tasks more challenging than GLUE, although I think this isn’t necessary for this work. However, the GPT-4 helpfulness scores seem promising.

Using GPT-4 for ratings is still a relatively new practice, and I think it is not yet well enough established that you can rely on it without having at least some testing of how well it correlates with human ratings - if you are not careful about things like positional bias this can result in results that may not correlate well with human judgement. At the very least, it would be useful to (a) check agreement with GPT-4 over a small subset of annotations, (b) compute the variation of GPT-4 scores over a set of different prompts to ensure no glaring biases.

Overall, I lean accept for this work, and think these weaknesses are mostly around clarifications, rather than weaknesses in the method. If the authors answer my questions well and make these clarifications, I am happy to accept.

Edit: I have raised my score after carefully reading the author's response and the other reviews. My issues have mostly been cleared up by rebuttal. Please see my response to the author rebuttal for more details.

**Questions:**

- How exactly is lambda chosen for each setting? What is your tuning / choice procedure?
- Why does it seem like lambda = 1.0 does best in Figure 3? Does this mean task composition is not useful in this case?
- In Figure 2, why do the performances on RTE/MNLI at lambda = 0/1 not match the values in Table 2?
- How well do GPT-4’s helpfulness scores correlate with human judgement?


**Limitations:**

The authors discuss their limitations to a reasonable extent.

---

> ### Author Rebuttal · Authors · 2023-08-09
>
> ### Q1: How $\lambda$ is chosen for each setting
> As mentioned in Line 137 of the submission, $\lambda$ is generally tuned on the validation set. Specifically, for classification tasks, we vary $\lambda$ from 0 to 1 in increments of 0.02 on a validation set with a limited amount of data. Similar $\lambda$ tuning when combining parameters is commonly practiced in [1], [2] and [3] as well. For negation settings in Section 4.4, we follow [4] and vary $\lambda$ from 1 to 0 with a step size 0.1 to find the maximum $\lambda$ to satisfy the requirement that the difference of PPL scores between the detoxified model  and GPT-2 baseline should not exceed 0.5 (as noted in Line 517 of the appendix) – we note that a detoxified validation dataset is not required here since PPL is computed on WikiText-103. For detoxifying LLaMA, $\lambda$ can often be determined based on one or two outputs –  we initially set $\lambda$ to 1 and generate sentences to evaluate their normality and linguistic proficiency manually. If the output is not satisfactory, we decrease the $\lambda$ value by 0.1 for the next iteration. Therefore, selecting the $\lambda$ value incurs minimal cost.
>
> We agree with the reviewer that the necessity of tuning $\lambda$ is certainly a limitation of our method when applied in practice, as we already acknowledged at the end of the submission. Further exploration on automatically reweighting parameters without tuning is an important research direction that we leave as future work.
> ### Q2: Do we train a module on the classification datasets?
> To clarify, In the domain transfer experiment, we did not train a module on the target classification dataset. The procedure is as follows: if we require θ_{yelp_cls}, but lack yelp classification data, and we have existing θ_{amazon_cls}, θ_{yelp_lm}, and θ_{amazon_lm}, we can obtain θ_{yelp_cls} by performing addition and negation operations among these three modules with the Equation in Line 232.
>
> ### Q3: $\lambda=1$ does the best in Figure 3?
> Actually $\lambda=1$ is not the best value in Figure 3 – when $\lambda$ approaches 1, the curve is becoming flat, making it difficult to distinguish which point is the optimal one. The optimal $\lambda$ values in Figure 3 and used in our experiment are as follows: in T5-base, 0.86 for Yelp and 0.98 for Amazon; in T5-small, 0.88 for Yelp and 0.96 for Amazon. As shown in the equation of Line 232, the classification module makes a more significant contribution, while the subtraction of language modeling modules remains crucial but less prominent in the domain transfer setting.
>
> ### Q4: In Figure 2, why do the performances on RTE/MNLI at $\lambda$ = 0/1 not match the values in Table 2?
> The numbers in Figure 2 and Table 2 are actually consistent: Figure 2 displays the variations in validation results of the composed LoRA for both MNLI and RTE tasks as $\lambda$ varies. For $\lambda$=0, the composed LoRA exhibits the same behavior as the LoRA trained on RTE, yielding 81.2 in RTE validation (red line in Figure 2, the leftmost point) and 54.7 in MNLI validation (blue line in Figure 2, the leftmost point). Conversely, for $\lambda$=1, the composed LoRA is equivalent to the LoRA trained on MNLI, achieving 75.8 in RTE validation (red line in Figure 2, the rightmost point) and 86.8 in MNLI validation (blue line in Figure 2, the rightmost point). These numbers match the ones in Table 2.
>
> ### Q5: How well do GPT-4’s helpfulness scores correlate with human judgement?
> Thank you for pointing out the potential bias issue of GPT-4 evaluation!  To mitigate the concerns on GPT-4 scoring and supplement more convincing results on the helpfulness evaluation,  we conduct pairwise human evaluation to compare the helpfulness of Alpaca-LoRA and our composed model. In our annotation scheme, we present the annotators the query and the responses from two models anonymously in a random order, and ask for a vote on the winner or tie.
> We begin our human evaluation by first assessing the inter-annotator agreement rate on a randomly sampled set of 50 examples out of 200 pairs of responses in the experiment. Three paper authors perform pairwise votes (model name is anonymized) and then we compute the agreement rate using tie-discounted accuracy following [5]. To compute the agreement rate between human and GPT-4, we convert our original GPT-4 scoring in the submission to pairwise votes similarly to [6]. As a result, the inter-annotator agreement rate of human-human is **78%**, human-GPT4 is **82%**, which are considered high and acceptable as observed in [5] and [6] as well.
>
> Subsequently, we acquired the results of the human evaluation for all 200 pairs of responses. The win rates of the detoxified merge module were 36% for toxic instructions and 27% for normal instructions with a 40% and 42% tie rate, indicating that the negation operation did not negatively impact the model's performance much in terms of helpfulness. These results are also consistent with our original GPT-4 scoring in the submission. We have shown the full results table including human eval in Response to Q2 of Reviewer wusG. We will add the human eval experiment to the next revision of the paper.
>
> [1] Matena, Michael S., and Colin A. Raffel. "Merging models with fisher-weighted averaging." NeurIPS. 2022.
>
> [2] Wortsman, Mitchell, et al. "Model soups: averaging weights of multiple fine-tuned models improves accuracy without increasing inference time." ICML. 2022.
>
> [3] Wortsman, Mitchell, et al. "Robust fine-tuning of zero-shot models." CVPR. 2022.
>
> [4] Ilharco, Gabriel, et al. "Editing models with task arithmetic." ICLR. 2022.
>
> [5] Zhou, Chunting, et al. "Lima: Less is more for alignment." arXiv preprint arXiv (2023).
>
> [6] Zheng, Lianmin, et al. "Judging LLM-as-a-judge with MT-Bench and Chatbot Arena." arXiv preprint arXiv (2023).

---

> > ### Comment · Reviewer_PsaT · 2023-08-14
> > **Re: Rebuttal**
> >
> > Hi, thanks for the detailed response! I realise now I found table 2 confusing due to the shared column/row names (and 'merge' being somewhat similar to 'avg' semantically) - it would be good to label the second column (detailing the trained module rte/mnli/merge) as something like 'module used' or 'module trained'. Your clarification clears up my confusion with table 2/figure 2, thanks. Similarly, thank you for the clarification with figure 3. It might be worth discussing the curve flattening in the paper text, as it's a useful insight that the subtraction of language modelling modules has a smaller effect in the domain transfer setting.
> >
> > I'm satisfied that the paper is novel and while the method is somewhat incremental, I think the experiments are solid, interesting, and useful for future work looking into module merging. So long as the authors do include the extra details and experiments given in rebuttals in the final version, I am happy to raise my score to accept.

---

> > > ### Author Response · Authors · 2023-08-15
> > >
> > > Thanks for your endorsement! In the next revision, we will make sure to include the details and experiments given in the rebuttal, and clarify table 2 as suggested.

---

### Official Review · Reviewer_wusG · 2023-07-06

**Soundness:** 3 good
**Presentation:** 3 good
**Contribution:** 2 fair
**Rating:** 7
**Confidence:** 3

**Summary:**

This paper proposes an efficient way to adapt pre-trained language models using parameter-efficient fine-tuning (PEFT). Instead of fully fine-tuning these models, the authors develop lightweight modules for each dataset, resulting in compact modules with varied skills. These modules are combined using linear arithmetic operations in the weight space, providing flexible module composition without needing extra training. This composition technique is applied to achieve distribution generalization, multi-tasking, detoxification, and domain transfer. The authors further extend this method to detoxify Alpaca-LoRA, a large language model. Empirical results suggest that this approach can create more effective modules that perform better than existing ones.

**Strengths:**

- the paper is scientifically sound and easy to read
- the idea of adapting pre-trained language models is interesting

**Weaknesses:**

- Without deviations and statistical tests, it is challenging to ascertain whether the model surpasses LoRA, given the close performance results.
- Utilizing GPT-4 for model evaluation could potentially introduce bias (e.g., see: https://arxiv.org/abs/2305.17493), as GPT-4 is inherently biased.
- The work appears to be an iteration of the paper from Pfeiffer et al., 2023, with restricted novelty.

**Questions:**

how would you emphasize the fundamental novelty of the paper comparing it with the one from Pfeiffer et al.?

**Limitations:**

Yes, the authors have sufficiently acknowledged the potential limitations of their work. They have considered the inherent biases or safety concerns that may be present in the Parameter-Efficient Modules (PEMs) they used. They've also touched upon the black-box nature of neural networks that might inadvertently introduce toxicity in certain scenarios.

They identified two main limitations: the restriction to identical PEM architectures and similar module initialization in most experiments, and the necessity of tuning the weight hyperparameter λ. They have also provided a direction for future work to address these limitations, which includes exploring different PEM architectures, varied module initialization, and automated computation of the weight hyperparameter.

Regarding potential negative societal impacts, the authors do not explicitly discuss this. However, they do allude to safety and bias issues inherent in the PEMs they utilize, which indirectly covers potential societal concerns. It would be beneficial for the authors to further discuss potential negative societal impacts explicitly in future work.

---

> ### Author Rebuttal · Authors · 2023-08-09
>
> ### Q1: Deviations and statistical tests for close results
> Thanks for the advice! In our submission, we only conducted statistical tests for the domain transfer experiment in Table 4.  We understand that some results in Table 1 are close and statistical tests are necessary there as well. We plan to set different random seeds and repeat the experiment multiple times to calculate the standard deviation. Due to the large number of experiments that need to be carried out for Table 1 and the limited time, we will include deviations for Table 1 in the next revision.
>
> ### Q2: Potential bias from using GPT-4 as the judge
> Thanks for the pointer and we agree that using GPT-4 as judge may incur biases. To mitigate concerns on this issue, we further run pairwise human evaluation to compare the helpfulness of *Alpaca-LoRA* and *Merge* in Table 5 of the detoxifying experiment.
>
> Specifically, we conducted a manual evaluation of a total of 200 pairs of responses from our experiment in Section 4.6, consisting of both toxic instructions and non-toxic instructions generated by the original Alpaca-LoRA and the detoxified merge module. The details of human evaluation are designed following LIMA [1] – we presented the annotators with two responses in random order and asked them to choose from three options: 'Model A wins', 'Model B wins', or 'Tie'. Initially, three evaluators, who are the authors themselves, assessed 50 of them to calculate their agreement rate using the tie-discounted accuracy following [1], which was found to be *78%*. A close-to-80% agreement rate is considered high and acceptable among human annotators, as practiced in LIMA, Chatbot Arena [2] and MT-bench [2]. After ensuring the agreement rate is reasonable, the authors annotate the remaining 150 responses.
>
> The results of the manual evaluation are shown in the following table.
> The win rate of the detoxified merge module are 36% for toxic instructions and 27% for normal ones with a 40% and 42% tie rate, which aligns with the observation from the original GPT-4 scoring in our submission. The results imply that the negation operation didn’t sacrifice the model performance significantly on helpfulness.
>
> | Method | Toxicity score $\downarrow$ |  | Toxic generation (\%) $\downarrow$ |  | Helpfulness score (GPT-4) $\uparrow$ |  | Helpfulness win/tie/lose rate (Human, \%)  |   |
> |---|---|---|---|---|---|---|---|---|
> |  | toxic | normal | toxic | normal | toxic | normal | toxic | normal  |
> | Alpaca-LoRA | 0.321 | 0.008 | 20 | 0 | 6.85 | 7.87 | 24/40/36 | 31/42/27  |
> | Merge | 0.158 | 0.008 | 6 | 0 | 7.13 | 7.63 | 36/40/24 | 27/42/31 |
>
> ### Q3: Novelty of the paper compared to Pfeiffer et al.
> The Pfeiffer et al.'s (2023) paper [3] mentioned by the reviewer is a comprehensive literature review of modular deep learning. Our study focuses on the problem classified under their 'Merging Modular Models' subsection of Section 9.1 Future Work. They neither explored nor conducted experiments on modular composition methods, thus our work is very different from theirs.
>
> A more related work to ours from Pfeiffer et al. is AdapterFusion [4]. We highlight the following key novelty of our paper compared to AdapterFusion:
> 1. The methodologies are essentially different – while we perform composition on the weight space and merge multiple PEMs into one PEM, AdapterFusion operates on top of outputs of PEMs and does not merge the parameters. Also, our method is based on the composition of addition/negation operators that we first proposed in this paper but did not exist in AdapterFusion.
> 2. AdapterFusion needs additional training on extra training data while our method is training-free.
> 3. AdapterFusion was proposed for multi-task settings and only focused on “adding” abilities of different modules. However, our paper designs both addition and negation operators and explores more diverse and flexible settings where AdapterFusion could not be trivially applied, such as the detoxifying and domain transfer settings.
>
> [1] Zhou, Chunting, et al. "Lima: Less is more for alignment." arXiv preprint arXiv (2023).
>
> [2] Zheng, Lianmin, et al. "Judging LLM-as-a-judge with MT-Bench and Chatbot Arena." arXiv preprint arXiv (2023).
>
> [3] Pfeiffer, Jonas, et al. "Modular deep learning." arXiv preprint arXiv (2023).
>
> [4] Pfeiffer, Jonas, et al. "AdapterFusion: Non-Destructive Task Composition for Transfer Learning." EACL. 2021.

---

> > ### Comment · Reviewer_wusG · 2023-08-14
> >
> > Thank you for your response, I've changed the rating to "Accept"

---

### Official Review · Reviewer_ZZbE · 2023-07-06

**Soundness:** 2 fair
**Presentation:** 3 good
**Contribution:** 3 good
**Rating:** 4
**Confidence:** 4

**Summary:**

This paper proposes an approach to compose parameter-efficient finetuning modules without requiring additional training. The modules can be added to combine capabilities, or negated to remove some abilities from the model. The paper shows how different combinations of modules may be used in multiple scenarios such as out-of-distribution generalization, multi-task learning, detoxification and domain transfer. The authors also show that their approach may detoxify an instruction-tuned language model.

**Strengths:**

While there are existing approaches to combine adaptors or parameter-efficient finetuning modules, the introduction of the negation operator allows for more complex operations.

The approach is evaluated in multiple scenarios (distribution generalization, multi-tasking, unlearning, domain transfer) and for two different types of parameter-efficient finetuning modules. It is generally reasonably effective.

The paper is fairly easy to follow. The experiments are mostly ordered in increasing order of complexity.

Being able to combine parameter-efficient finetuning modules without additional training allows for a cheap and flexible mechanism to adapt large language models.

**Weaknesses:**

The main weakness of the paper (in my opinion) is the lack of comparison to existing approaches (e.g. Pfeiffer et al. 2021, Wang et al. 2022 (already cited)), especially for tasks where only the addition operator is needed. While the proposed approach is simpler than those requiring additional training, it is unclear whether performance is worse, comparable or superior.

For multi-task experiments, there is a noticeable drop in performance for RTE. While this may not be surprising, this is still somewhat concerning, especially given the limited comparison to other approaches.

[Minor] Given the weight $\lambda$, the addition operator is actually a weighted average operator.

**Questions:**

How does the proposed approach compare to existing work combining PEFT modules (even if some of them may not use LoRA or (IA)$^3$ directly)?

**Limitations:**

Yes, the authors have addressed limitations of their work.

---

> ### Author Rebuttal · Authors · 2023-08-09
>
> ### Q1: Comparison to other PEM combining work
> This is a good point. There have indeed been some works that combine multiple PEMs in the past, such as Pfeiffer et al. 2021 [1] and Wang et al. 2022 [2] mentioned by the reviewer. However, these approaches are not comparable to ours because (1)  they require additional training and access to training data, and (2) they were only demonstrated on “addition” settings to combine abilities and could not be trivially applied to our settings that involve negation operations, such as detoxification. This is why we did not include them in our experiments. Even so, we agree with the reviewer that it may be helpful to include their results as well to understand the potential performance gap. Therefore, we run AdapterFusion [1] in our multi-task setting and show the results below.
>
> | Method |  | RTE | MNLI | Avg.  |
> |---|---|---|---|---|
> | LoRA | AdapterFusion | 84.8 | 86.2 | 85.5  |
> |  | RTE | 81.2 | 54.7 | 68.0  |
> |  | MNLI | 75.8 | 86.8 | 81.3  |
> |  | Merge | 78.7 | 86.3 | 82.5  |
>
> We utilized the same LoRA modules employed in our PEM composition as the base PEMs in AdapterFusion. Then, we train AdapterFusion using the combined RTE and MNLI training datasets in a standard multi-task training setting as described in [1]. The training hyperparameters follow the ones in [1]. As shown in the table, AdapterFusion method yielded improvements with an RTE accuracy of 84.8 and an MNLI accuracy of 86.2. AdapterFusion outperforms our method by 3.0 points in terms of the average performance of MNLI and RTE, which is expected due to the extra multi-task training. We would like to emphasize again that these previous work like AdapterFusion require training and access to the respective training data, thus their results are not directly comparable and can only serve as a reference point. In future revisions, we will add AdapterFusion results in our other settings where only the addition operator is needed (i.e. the distribution generalization and the multi-tasking settings).
>
> ### Q2: Noticeable drop in RTE of multi-task experiments
> Despite a slight drop in performance on each individual task, we would like to note that the combined model performs well across multiple tasks, which can be viewed as an advantage over a single model that excels in only one task. Moreover, such single-task performance drop is also commonly observed in previous work on model merging [3,4,5]. Therefore, we think that although the performance drop on a single task in multi-task settings is not ideal, it is understandable at this stage. More advanced composition methods to fill this gap are left as future work.
>
> [1] Pfeiffer, Jonas, et al. "AdapterFusion: Non-Destructive Task Composition for Transfer Learning." EACL. 2021.
>
> [2] Wang, Yaqing, et al. "AdaMix: Mixture-of-Adaptations for Parameter-efficient Model Tuning." EMNLP. 2022
>
> [3] Jin, Xisen, et al. "Dataless Knowledge Fusion by Merging Weights of Language Models." ICLR. 2022.
>
> [4] Qin, Yujia, et al. "Exploring Mode Connectivity for Pre-trained Language Models." EMNLP. 2022.
>
> [5] Ilharco, Gabriel, et al. "Editing models with task arithmetic." ICLR. 2022.

---

> > ### Comment · Reviewer_ZZbE · 2023-08-21
> >
> > Thank you for your response and for reporting additional results.
> >
> > For Q1, I agree that these other methods have a distinct advantage by using training data, so the results may not be directly comparable. I would suggest to more clearly demonstrate that "the corresponding training data is often unavailable" (from the intro).

---

> > > ### Author Response · Authors · 2023-08-21
> > >
> > > Thanks for the suggestion! We will add clarification on this to the intro in the next revision. Since the original review says that "the main weakness of the paper is the lack of comparison to existing approaches" and we have reached consensus after the rebuttal that the previously mentioned approaches may not be directly comparable to our method, does it change your mind on the original review rating?

---

> > > > ### Comment · Reviewer_ZZbE · 2023-08-21
> > > >
> > > > I am considering increasing my rating, but I first need to take a closer look at the concerns raised by reviewer hxSN.
> > > >
> > > > As mentioned above, it is important for readers to understand when the other methods are not applicable. The drop in RTE is unfortunate, although it won't block my recommendation of the paper.

---

### Official Review · Reviewer_EMdb · 2023-07-10

**Soundness:** 3 good
**Presentation:** 3 good
**Contribution:** 2 fair
**Rating:** 7
**Confidence:** 5

**Summary:**

The authors proposed to perform an arithmetic combination of PEFT Modules. Suggested combinations were evaluated on distribution generalization, multitasking, detoxifying, and domain transfer tasks. Authors showed that combining PEFT Modules produces new modules with desired attributes.

**Strengths:**

- The proposed method is interesting for practitioners
- Experiments are mainly well designed
- The paper is well-written and easy to follow

**Weaknesses:**

- For Table 1, it would also be highly beneficial to include fine-tuning results on the full dataset to understand how merging modules compares to it.
- It would be interesting to see any ablations on the design choices of the PEM Negation operator. Claims that one could not naively negotiate weights of LoRA (L120) seem unsupported by any evidence. Analysis of results with different types of negotiation would make the paper better. Furthermore, FFT negotiation from Section 4.4 implied such naive negotiation of all weights.
- While the authors explored a wide range of different tasks to understand the performance of the proposed approach, I found that the paper needed an in-depth analysis of the results. E.g., when speaking of detoxifying (Section 4.4), the only available results are the final toxicity score and PPL of obtained model. Though, there are more automatic metrics for text generation, such as a number of distinct n-grams, which add more dimensions to understanding performance. Furthermore, while discussing detoxifying, there are many fascinating things to do with negotiation. E.g., for FFT, it is explored to perform negotiation with a weight larger than $1$ (https://arxiv.org/abs/2211.12092).

**Questions:**

Please, refer to the weaknesses section

**Limitations:**

–

---

> ### Author Rebuttal · Authors · 2023-08-09
>
> ### Q1: Fine-tuning results on the full dataset for Table 1
> Thanks for your advice! We run LoRA-tuning on the combination of the two subsets s0 and s1 for the eight GLUE tasks described in Table 1, and show the results in the following table (denoted as “full dataset”). Not surprisingly, we observe that merging modules slightly underperforms fine-tuning on the full dataset, implying that more advanced composition methods are worth further exploration to improve the performance. However, we note that full-dataset tuning requires access to both subsets during training and represents a different setting from our merging experiments, thus its numbers should be viewed as a reference point only. We will add the full-dataset tuning results into Table 1 for completeness in future revisions.
>
> | Method |  | MNLI | RTE | SST-2 | MRPC | QNLI | QQP | CoLA | STS-B  |
> |---|---|---|---|---|---|---|---|---|---|
> | LoRA | full dataset | 76.8 | 76.5 | 92.8 | 88.0 | 84.8 | 80.6 | 0.55 | 0.90  |
> |  | $s_0$ | 71.4 | 72.2 | 92.2 | 86.3 | 83.1 | 79.0 | 0.50 | 0.88  |
> |  | $s_1$ | 72.3 | 69.0 | 91.9 | 87.7 | 83.0 | 80.8 | 0.51 | 0.89  |
> |  | $m$ | 73.5 | 75.8 | 92.2 | 88.0 | 83.3 | 81.1 | 0.52 | 0.89 |
>
> ### Q2: Ablation analysis of the PEM negation operator
> Thanks for your advice!  We conduct ablation experiments on LoRA and $\rm{(IA)^3}$ negation in the setting of Section 4.4. The following table presents the results where we naively negate all weights of the modules (denoted as “weight-negated”).  For LoRA, the results after naively negating are identical to the original toxic LoRA because they are theoretically equivalent as we described in Line 120 of the submission. As for $\rm{(IA)^3}$, when naively negating the scaling vector $l$, it causes a catastrophic performance drop. When we vary the value of $\lambda$ in the range of 0.1 to 1, we could not find an appropriate $\lambda$ where the model generates fluent text – in contrast, the model always produces incomprehensible output that results in a high PPL. As a consequence, the toxicity score and metrics related to toxic generation become meaningless with a high PPL. These results demonstrate that negating the weights of PEMs naively is not effective for PEM composition.
>
> | Method | Toxicity score $\downarrow$ | Toxic generation (\%) $\downarrow$ | PPL $\downarrow$ | Distinct n-grams $\uparrow$ |
> |---|---|---|---|---|
> | GPT-2 | 0.10 | 5.8 | 16.44 | 0.467  |
> | toxic-FFT | 0.59 | 50.2 | 16.46 | 0.509  |
> | toxic-LoRA | 0.43 | 34.3 | 17.00 | 0.474  |
> | toxic-$\rm{(IA)^3}$ | 0.26 | 20.5 | 17.33 | 0.510  |
> | negated-FFT $(\lambda=0.5)$ | 0.04 | 2 | 16.94 | 0.490  |
> | negated-LoRA $(\lambda=1)$ | 0.01 | 0.1 | 16.67 | 0.467  |
> | negated-$\rm{(IA)^3}$ $(\lambda=0.6)$ | 0.03 | 0.9 | 16.92 | 0.488  |
> | weight-negated-LoRA $(\lambda=1)$ | 0.43 | 34.3 | 17.00 | 0.474  |
> | weight-negated-$\rm{(IA)^3}$ $(\lambda=1)$ | 0.11 | 8.7 | 843.83 | 0.265  |
> | weight-negated-$\rm{(IA)^3}$ $(\lambda=0.6)$ | 0 | 0 | 5.91E+04 | 0.021  |
> | weight-negated-$\rm{(IA)^3}$ $(\lambda=0.1)$ | 0 | 0 | 3.00E+09 | 0.008 |
>
> ### Q3: More automatic metrics for text generation in detoxifying
> Thanks for the advice! We use PPL and toxicity scores mainly following the setting in [1]. We agree with the reviewer's suggestion and introduce distinct n-grams (n=1) as an additional measure for assessing the diversity of the generated text. We use the Distinct method from PaddleNLP and the table above presents the results. Our analysis reveals that incorporating negation does not diminish the n-grams score compared to the GPT-2 baseline, which implies that the diversity of the generated text remains consistent with the original model.
>
> ### Q4: There are many fascinating things to do with negotiation for detoxifying
> Thanks for the pointers and we agree! Our paper is mainly to study the general effect of composition in various settings, and detoxifying is just one of them that we take as an example. Thus, we only explored relatively simple variants in all our experiments to keep the paper concise, and we leave further study on more composition variants in different settings as future work.
>
> [1] Ilharco, Gabriel, et al. "Editing models with task arithmetic." ICLR. 2022.

---

> > ### Comment · Reviewer_EMdb · 2023-08-10
> >
> > Thank you for your detailed response. I'm more than satisfied with it.

---

> > > ### Author Response · Authors · 2023-08-11
> > >
> > > We are glad that you are satisfied with our response! If our response (partially) addressed your concerns, would you like to consider updating the review rating accordingly?

---

### Author Rebuttal · Authors · 2023-08-09

We thank the reviewer for the time and comments, and we reply to the comments of each reviewer separately in the respective thread. Due to time limitations we could only address major points, but we’ll make sure to reflect all advice in future revisions.

---

### Decision · Program_Chairs · 2023-09-21

**Decision:**

Accept (poster)

**Comment:**

The researchers proposed a novel approach involving the arithmetic combination of PEFT Modules. The suggested combinations underwent evaluation across various tasks including distribution generalization, multitasking, detoxification, and domain transfer. The study demonstrated that combining PEFT Modules produces new modules with desired attributes.

After the rebuttal, there is only one reviewer who did not change the negative review. This reviewer raised a novelty issue about the negation operators, while the authors shown their unique contribution as the first to explore both addition and negation operators for PEM composition.

Given that the rebuttal effectively addressed the majority of concerns raised by the reviewers, I am pleased to recommend the acceptance of this paper. I look forward to seeing the authors incorporate the necessary revisions into the final version for camera-ready submission.